# Improving Upper Extremity Bradykinesia in Parkinson’s Disease: A Randomized Clinical Trial on the Use of Gravity-Supporting Exoskeletons

**DOI:** 10.3390/jcm11092543

**Published:** 2022-05-01

**Authors:** Loredana Raciti, Loris Pignolo, Valentina Perini, Massimo Pullia, Bruno Porcari, Desiree Latella, Marco Isgrò, Antonino Naro, Rocco Salvatore Calabrò

**Affiliations:** 1GCA-Centro Spoke AO Cannizzaro, Catania, IRCCS Centro Neurolesi Bonino-Pulejo, 98124 Messina, Italy; loredana.raciti@irccsme.it; 2S. Anna Institute, Research in Advanced Neurorehabilitation, 88900 Crotone, Italy; lpignolo@gmail.com; 3Spoke Centre of Palermo, IRCCS Centro Neurolesi Bonino-Pulejo, 98124 Messina, Italy; valentina.pirini@irccsme.it; 4Behavioral and Robotic Neurorehabilitation Unit, IRCCS Centro Neurolesi Bonino Pulejo, 98124 Messina, Italy; massimo.pullia@irccsme.it (M.P.); bruno.porcari@irccsme.it (B.P.); desiree.latella@irccsme.it (D.L.); marco.isgro@irccsme.it (M.I.); 5Department of Clinical and Experimental Medicine, University of Messina, 98122 Messina, Italy; g.naro11@alice.it

**Keywords:** gravity-supporting device, hand bradykinesia, Parkinson’s disease, upper-limb rehabilitation, neurodegenerative diseases

## Abstract

Hand movements are particularly impaired in patients with Parkinson’s Disease (PD), contributing to functional disability and difficulties in activities of daily living. Growing evidence has shown that robot-assisted therapy may be considered an effective and reliable method for the delivery of the highly repetitive training that is needed to trigger neuroplasticity, as intensive, repetitive and task-oriented training could be an ideal strategy to facilitate the relearning of motor function and to minimize motor deficit. The purpose of this study is to evaluate the improvement of hand function with semi-autonomous exercises using an upper extremity exoskeleton in patients with PD. A multicenter, parallel-group, randomized clinical trial was then carried out at the IRCCS Centro Neurolesi Bonino-Pulejo (Messina, Italy). Thirty subjects with a diagnosis of PD and a Hoehn–Yahr score between 2 and 3 were enrolled in the study. Patients were 1:1 randomized into either the experimental group (ERT), receiving 45 min training daily, 6 days weekly, for 8 weeks with Armeo^®^Spring (Volketswil, Switzerland) (a gravity-supporting device), or the control group (CPT), which was subjected to the same amount of conventional physical therapy. Motor abilities were assessed before and after the end of the training. The main outcomes measures were the Nine-hole peg test and the motor section of the UPDRS. All patients belonging to ERT and 9 out of 15 patients belonging to the CPT completed the trial. ERT showed a greater improvement in the primary outcome measure (nine-hole peg test) than CPT. Moreover, a statistically significant improvement was found in ERT concerning upper limb mobility, and disease burden as compared to CPT. Using an upper extremity exoskeleton (i.e., the Armeo^®^Spring) for semi-autonomous training in an inpatient setting is a new perspective to train patients with PD to improve their dexterity, executive function and, potentially, quality of life.

## 1. Introduction

Bradykinesia and hypo-akinesia are the clinical hallmarks of Parkinson’s disease (PD), mostly impairing the sequentiality and dexterity of upper extremities (UE) movements [1]. Notably, the motor impairment degree in PD patients has been shown to inversely correlate with movement velocity and directly with task difficulty [2,3]. Indeed, UE functionality impairment is usually characterized by aberrant timing and force modulation, resulting in a poor quality of hand movements [4]. These alterations contribute to the impairments of body functions and structures and the difficulty in activities and participation [5,6], leading to a lack of independence and poor quality of life [7,8].

Whether motor control deficits in PD are related to the slowness of movements rather than to impairments in movement selection is still controversial, as the nature of bradykinesia is still under investigation [9,10]. Initially, some authors suggested that bradykinesia is due to an inability to generate appropriate muscle activity before and during voluntary movements for the completion of the intended movement [11]. This was likely related to executive function deficit and reduced motor information-processing in PD [11], resulting in hypometric and increasingly impaired movements when the difficulty of a sequential motor task increases [11]. Additionally, a decreased ability in thumb–index maximum torque generation was found in PD patients [12]. Moreover, the basal ganglia have been recently described to have a pivotal role in the pathogenesis of the disorder. Then, PD is considered the consequence of the disruption of the excitability and plasticity of the basal ganglia circuits, specifically of the M1 and other non-primary motor areas [9,13].

Some therapies have proven to be effective in the improvement of motor function, enhancing the impact of non-motor symptoms and improving quality of life, even though a poor response has been demonstrated concerning hand movement dexterity [14]. To date, only a few studies focused on intervention strategies on UE and hand movements, particularly regarding fully adaptive, assist-as-needed exercises of the traditional rehabilitation [15].

Indeed, most of the existing data have focused on gait using either robotics [16] or other less expensive devices to improve the different gait parameters in patients with PD [17].

Innovative technologies are widely used in many neurological diseases, including stroke, multiple sclerosis, and spinal cord injury, with the paramount effort to increase the active range of motion and muscle strength [18,19]. Moreover, it has been shown that better outcomes could be achieved when robotics is coupled to virtual reality, resulting in a potentiation of body function and activities [20]. Such promising results depend on the fact that conjugating robotics and virtual reality allows maximizing neural, motor and functional recovery, by providing patients with lasting, challenging, repetitive, task-oriented, motivating, salient, and intensive motor exercises; furthermore, virtual reality may also provide patients with a potentiation of motor and cognitive functions [20,21,22,23].

There is an increasing interest in robotic rehabilitation in PD. Some studies showed that the use of robot-assisted rehab is useful for gait training [15,24], improve freezing of gait [25] to reduce the muscular activity requirements of PD patients [26], balance [27], and reduce the amplitude of tremors [28]. Then, the use of advanced and computerized technologies could provide a better rehabilitation in patients with PD [29,30]. However, no data are available about the use of specific robotic devices to recover arm impairment in PD patients. 

The aim of this pilot study is to evaluate the efficacy of a gravity-supporting exoskeleton apparatus (i.e., the Armeo Spring; Hocoma, Zurich, Switzerland) on hand dexterity and overall motor functions in PD patients (experimental robotic therapy-ERT) as compared to conventional physical therapy (CPT).

## 2. Materials and Methods

### 2.1. Study Design and Population

We carried out a parallel-group, single-blinded, randomized controlled trial on the PD patients attending either the Movement Disorders Clinic of the IRCCS Centro Neurolesi Bonino-Pulejo (Messina, Italy) or its Spoke Centre (Palermo, Italy) between July 2019 and March 2020. Indeed, the enrollment period was interrupted due to the COVID-19 pandemic and some patients abandoned the study (Figure 1).

The inclusion criteria were as follows: (i) history of idiopathic PD diagnosed according to the UK Brain Bank criteria [31]; (ii) a Hoehn–Yahr stage between 2 and 3 determined in the “on” phase [32]; and (iii) age between 50 and 80 years old. The exclusion criteria were: (1) moderate to severe cognitive deficits potentially limiting comprehension of the experimental task (i.e., a Mini-Mental State Examination < 20) [33]; (2) severe dyskinesia or severe on–off motor fluctuations; (3) stereotaxic brain surgery for PD; (4) changes in dopamine therapy dose (as per levodopa equivalent daily dose, LEDD) within 3 months prior to baseline; (5) unstable cardiac or respiratory illness potentially interfering with the training; and (6) any other medical condition that could compromise the training, including severe osteoarthritis or peripheral neuropathies.

Written informed consent was obtained from all participants. The study was conducted according to the laws, regulations, and administrative provisions relating to the implementation of good clinical practice in the conduct of clinical trials on medicinal products for human use, as applicable by national legislation and the Declaration of Helsinki. The protocol (No. U0074917/11110) was approved by the Ethical Committee of the IRCCS Centro Neurolesi Bonino Pulejo (Messina, Italy) and was registered at ClinicalTrials.gov (NCT02721212).

### 2.2. Randomization

After the screening session, when patients satisfied all inclusion criteria, they were randomly assigned to ERT or CPT. To this end, we adopted a computerized randomization stratification approach. The randomization scheme (1:1 ratio) was set up in permuted blocks of three to ensure a similar number of participants between groups. Additionally, participants were stratified by their degree of disability and impairment (Unified Parkinson’s Disease Rating Scale score, UPDRS) [34] to obtain a balance between groups regarding the baseline physical capacity. The assessors were blinded to the group allocation of participants.

### 2.3. Sample Size and Power Analysis

The sample size, in relation to the clinically significant changes in the primary outcome, was calculated by means of a 2-sided, 2-sample *t*-test, and estimated in 60 patients (30 per arm). More specifically, this sample size was required to maintain a type I error rate of 0.05 and an 80% power to detect a significant between-group difference of 5.65 s [35] (including a 10% dropout rate or loss to follow-up).

### 2.4. Experimental Robotic Therapy

The Armeo^®^Spring (Figure 2) is a mechanical device (Hocoma Inc., Zurich, Switzerland) characterized by an adaptable suspension system for the upper limb that offers support from the shoulder to the wrist ending with a grasping system for the hand.

System sensitivity can be adjusted depending on the patient’s condition. The system gives information about movement parameters, such as resistance, strength, range of motion, and coordination. Based on the patient’s active mobility, the system allows to calibrate the working space and level of difficulty during the entire training period. The device expands any movement, allowing reinforcement and facilitation of the arm by visual feedback (2D virtual reality) with a three-dimensional space [36].

The training was supervised by a physiotherapist expert in robot-assisted therapy. After the device was adjusted for the patient’s arm size and angle of suspension, the workspace and exercises were selected. Exercise difficulties were modified during the following sessions. Each training session consisted of 45 min per session for each arm with Armeo^®^Spring, 6 days per week, for 8 weeks.

### 2.5. Conventional Physical Therapy

Patients in the control group received the same global treatment time of the ERT. Patients were submitted to conventional rehabilitation, such as passive- and active-assisted mobilization of the upper limbs, traditional training for neuromuscular facilitation, proprioception exercises, and reducing joints and muscles stiffness. Active exercises of reaching and picking objects were also performed.

### 2.6. Outcome Measures

We provided the patients with the motor section of the UPDRS and the Motricity Index for Upper Extremity (MI-UE) to test motor impairment, the nine-hole peg test (9HPT) [37] to objectively evaluate hand dexterity, the Fugl-Meyer Assessment for the upper extremity (FMA-UE) to test the UE motor ability to perform selective movements, the Functional Independence Measure (FIM) to estimate the disability burden, and the numerical rating scale of pain (P-NRS) to measure the range of pain intensity. The primary outcome was the change of the 9HPT from baseline (T0) to post-treatment (T1). The secondary outcomes were the changes of UPDRS, FMA-UE, FIM, P-NRS, and MI-UE from T0 to T1.

#### 2.6.1. Primary Outcome Measure

The 9HPT test was used to assess hand dexterity, by asking the patients to take nine pegs from a container and place them into nine holes on a board and vice versa as quickly as possible. Score was the time taken to complete the test activity [38].

#### 2.6.2. Secondary Outcome Measures

The UPDRS is the most used screening tool to detect disease severity and motor and non-motor complications in PD [34,38]. It consists of three subscales: part I: evaluation of mentation, behavior, and mood by an interview; part II: the activities of daily life by self-evaluation; part III or motor section is a clinician-scored motor evaluation; and part IV is the evaluation of long-term levodopa complications, such as dyskinesia and motor fluctuations.

The FMA-UE was used to evaluate upper limb motor ability to perform selective movements. Of the motricity scales, the most popular used scale in PD is the Fugl-Meyer Motor Assessment Scale [39,40].

The 33-item scale consists of 3 response categories (scores 0–2) for each item, with a maximum score of 66 (indicating no impairment) with sub-scores of 36 for the upper arm, 10 for the wrist, 14 for the hand, and 6 for coordination and speed of movement [39,41].

The FIM measures the level of a patient’s disability burden and indicates how much assistance is required for the individual to carry out activities of daily living. Each task is rated on a 7-point ordinal scale that ranges from 1 = total assistance (or complete dependence) to 7 = complete independence [42].

The P-NRS typically consists of a series of numbers with verbal anchors representing the entire possible range of pain intensity. Generally, patients rate their pain from 0 to 10, from 0 to 20, or from 0 to 100. Zero represents “no pain”, whereas 10, 20, or 100 represent the opposite end of the pain continuum (e.g., “the most intense pain imaginable”, “pain as intense as it could be”, and “maximum pain”) [43].

The MI-UE can be used to assess motor impairment. MI is a feasible measure that can demonstrate the overall patients’ impairment. It is a simple, brief measure of general motor function that can predict mobility outcomes post-stroke [44]. All motor assessment and motor training sessions were performed bilaterally, in the affected and unaffected side.

### 2.7. Statistical Analysis

Between-group T0 differences were estimated using the Mann–Whitney test. The significance of the changes in each outcome measure was calculated by conducting a repeated-measure ANOVA for continuous numerical variables (using time, two levels, and group, two levels, as factors) or a Wilcoxon signed-rank test for ordinal or nominal variables. Depending on the significance of the main interactions and effects, Bonferroni corrected pairwise comparisons using *t*-tests, Wilcoxon test, or Mann–Whitney test, where appropriate, were tested. For all statistical tests, the significance level was set at α < 0.05. All the analyses included all participants for which data were available.

## 3. Results

### 3.1. Baseline (T0)

Thirty-three patients were consecutively screened for study inclusion. Three failed the inclusion criteria. Therefore, 30 patients were randomized to ERT or CPT. There were no differences at baseline between the groups concerning age, disease duration, and LEDD (Table 1; all *p* > 0.2).

Particularly, both groups had a Hoehn–Yahr stage between 2 and 3, which corresponded to an on-average moderate impairment of body functions and structure and a moderate difficulty impairment in activities and participation (i.e., present <50% of time, with intensity that interferes with day-to-day lift, occurring occasionally over last 30 days). The 9HPT, MI-UE, and FIM data were not available for three subjects belonging to the ERT group (Table 2 and Table 3).

### 3.2. Post-Treatment (T1)

Six patients belonging to the CPT dropped out (Figure 1). Therefore, 15 patients belonging to ERT and 9 to CPT were analyzed. No adverse events were reported, apart from the worsening of Pisa Syndrome in one patient. The analyses showed that ERT achieved a greater improvement in 9HPT than CPT (Table 2). Furthermore, ERT showed a greater improvement in UPDRS III, MI-UE, and FMA-UE (Table 2). On the other hand, both ERT and CPT equally improved in P-NRS (Table 2). Most of the patients reported an improvement in FIM; for instance, a patient started to work at the crochet again. When considering the most affected UE (Table 3), greater changes were appreciable concerning P-NRS, MI-UE, and FIM in ERT than in CPT, whereas both groups equally improved in 9HPT and FMA-UE.

## 4. Discussion

The PD patients included in the study complained of altered UE functions (including impaired timing and force modulation), with a compromised quality of hand movement. Furthermore, all patients complained of a mild to moderate motor impairment and ability to perform selective movements (UPDRS, MI-UE, and FMA-UE), disability burden (FIM), and very mild pain rating (P-NRS). Overall, they thus complained of a moderate impairment of body functions and structures and a moderate difficulty in activities and participation consistently with the International Classification of Functioning, Disability and Health [6].

To the best of our knowledge, this is the first study preliminarily evaluating the efficacy of an exoskeleton-based rehabilitation strategy in potentiating UE functions in PD patients as compared to CPT [4]. Actually, there are not available specific rehabilitation protocols or approaches to improve UE dexterity learning in PD. Both rehabilitation approaches were able to provide PD patients with a significant improvement in UE functions. However, ERT offered patients with specific benefits in terms of hand dexterity, overall UE movement, and disability burden as compared to CPT.

This finding is consistent with a pilot study by Picelli et al. [14], which showed the efficacy of the Bi-Manu-Track (Reha-Stim, Berlin, Germany) training in PD patients. In particular, ten sessions of bilateral (mirror-like) passive and active, computer-controlled, repetitive practice of forearm pronation/supination and wrist extension/flexion improved 9HPT and the FMA-UE. Despite the results of our study and those by Picelli et al. not being directly comparable due to the different devices used, both studies suggest that repetitive, robot-assisted UE training would be a promising tool to improve motor outcomes in PD patients. Furthermore, our results are in line with those by Lee et al., who observed significant improvements of UE fine and gross motor performance after several, repetitive treatment sessions by constraint-induced movement therapy in twenty patients with PD [32].

The superiority of gravity-supporting device as compared to CPT in reducing the impairment of, at least, some motor outcomes in PD patients is likely to depend on the intensive, repetitive, assisted-as-needed, and task-oriented motor practice provided by the mechanical device [45]. It has been proposed that such an approach allows exploiting the brain neural plasticity mechanisms both within affected and intact brain structures through the stimulation of motor learning processes and a reshape of the “inter-hemispheric inhibition” mechanisms by means of repetitive sessions of robot-assisted therapy, as demonstrated by UE dexterity improvement in post-stroke patients [46,47,48]. This extends to the case of basal ganglia impairment, where an intensive, continual, and contextual training allows facilitating the relearning of motor functions and minimizing motor deficit by acting on the internal regulator mechanisms of movement flow and programming [17,45,49]. Particularly, the improvement in motor outcomes following ERT sessions could be considered the expression of a relearning process resulting from the stimulation and the activation of the mirror neuron system, inducing profound cortical and subcortical changes at both the cellular and synaptic levels [50]. The role of the cerebellum in motor and reinforcement learning has been recognized. Recently, it has been shown that the cerebellum may have a compensatory and adaptive role concerning gait function recovery by favoring the precise timing of motor actions along the gait cycle phases. This probably occurs by compensating the deficient internal timing clock within the basal ganglia [51]. These results have been obtained in PD patients receiving gait training plus music (i.e., a repetitive exercise with an external cue). It is then hypothesizable that UL robotic rehabilitation with VR may have boosted neural plasticity also at the cerebellar level, further improving motor recovery.

In addition, UE training benefitted from the adjunction of virtual reality treatment. This is thought to increase the motor learning of well-defined motor tasks and improve motivation due to the feedback provided by the device itself [18,19,20,21,22,23,48,49,50,52,53,54]. Overall, these motor improvements allowed the patients to achieve a significant improvement in UE functions and disability burden. For instance, a patient was able to do a crochet work carried out after a 25% of ERT sessions. Such activity was referred to as difficult to carry out for about two years by the patient.

Although we have not specifically investigated quality of life, it is possible that the improvement in disability burden has also improved patients’ health-related quality of life, as suggested by other chronic diseases [55].

### Strengths and Limitations

Our data suggest that UE training using a gravity-supporting device could be considered a safe and effective approach for the recovery of UE motor functions in PD patients. Additionally, the rehabilitation technology is a cost-effective practice to reduce the need for one-to-one skilled interventions. The training sessions can be performed with minimal supervision once the therapist has set up the exercises. The attention of the patient is triggered by the visual and acoustic feedback while the exercise is performed. Moreover, the training is reproducible and essentially safe and can be easily conducted in the hospital setting. However, our study is limited by the small sample size (we were able to enroll only 30 patients compared to the estimated 60 because of the onset of COVID-19 pandemic). This could have had relevant consequences on the results, considering that an intention-to-treat analysis was not performed. It is true that an intention-to-treat approach tends to under-estimate an effect, being thus a more conservative approach in a clinical superiority trial. Nevertheless, a per-protocol approach is a reasonable analysis strategy for sensitivity analyses. Actually, a per-protocol approach is suitable when the exclusion of patients from the analysis due to major protocol deviations (which can of course cause a tendency to wrong estimations of a treatment effect) tends to vary among the study groups. However, it is not straightforward to pre-guess the direction of a wrong estimation and the general claim that a per-protocol analysis tends to overestimate an effect that cannot be mathematically proven [56,57,58]. Studies with larger and homogeneous samples could allow for a more accurate statistical analysis, including multivariate data analysis or more advanced tools, such as machine learning, aimed to point out predictive marker of recovery.

Another main limitation of our study is the lack of evaluation of long-term efficacy of the ERT (also due to COVID-19 pandemic restrictions). Nonetheless, previous studies have shown a persistent efficacy of robotic rehabilitation after two weeks only for the 9HPT [15]. Moreover, Taveggia et al. recorded a stability of motor assessment after 6 weeks from the end of treatment in stroke patients [59]; lastly, the improvement in the functional capacity outcome measures were found at 2-month follow-up in multiple sclerosis patients [60]. Last, one could criticize that the use of MI-UE is not usual in evaluating clinical impairment in PD, compared to the FMA-UE [61]. In fact, MI-UE assesses the motor impairment in a patient who has had a stroke evaluating the ability of the patient to hold a cube against gravity, the capacity to flex the elbow from 90° (so that the arm touches the shoulder) and the capacity of the shoulder to abduct to more than 90° beyond the horizontal against gravity. Rigidity in PD refers to increased muscular tone with more resistance than normal when the limb is passively moved, experienced by the patients as a sense of feeling stiff and uncomfortable. Then, even though the physiopathology of abnormal movements in PD are different from strokes [59], similar difficulties in achieving MI-UE goals are seen in PD patients. Then, further validation studies are needed to confirm the possibility to use the scale even in Parkinsonism. Larger sample size RCTs are, notwithstanding, needed to address this concern.

## 5. Conclusions

Our data suggest that exoskeleton-assisted therapy, such as the Armeo©Spring, may represent a safe and effective strategy for delivering a highly intensive and repetitive training, which is necessary to trigger the neuroplasticity mechanisms subtending UE motor function improvement. Few studies are however available on the UE rehabilitation of patients with PD, particularly regarding technology-enhanced physical therapy, except for gait training. Therefore, further investigations with larger sample sizes are needed to confirm our results and to optimize PD-specific rehabilitation protocols.

## Figures and Tables

**Figure 1 jcm-11-02543-f001:**
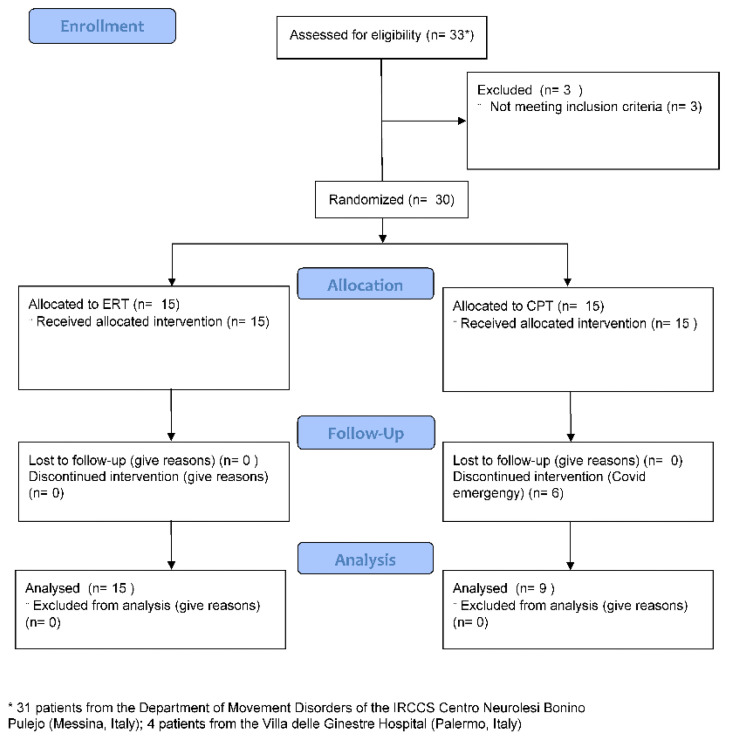
Patients’ diagram flow.

**Figure 2 jcm-11-02543-f002:**
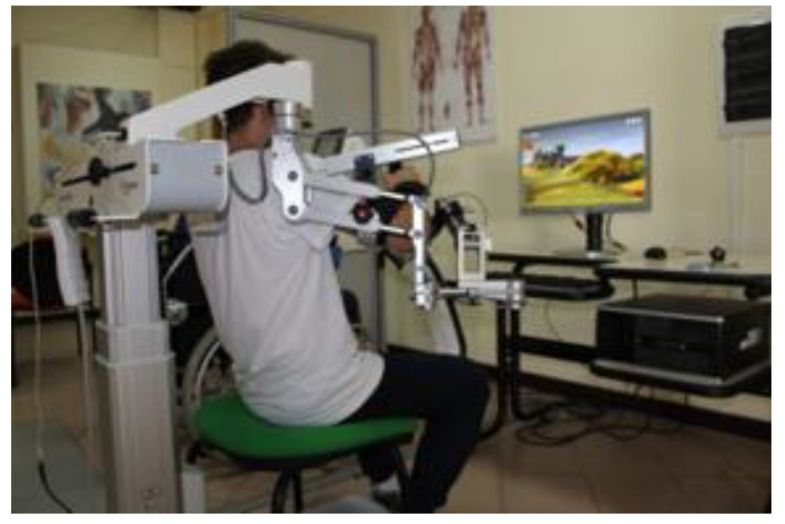
A patient with PD receiving the experimental training with the Armeo©Spring device.

**Table 1 jcm-11-02543-t001:** Clinical-demographic features at baseline (T0). Data are reported as mean (SD) or median (interquartile range), and *p*-value of Mann–Whitney test.

	ERT (*n* = 15)	CPT (*n* = 9)	*p*-Value
Age (years)	65.7 (7)	62.7 (10.1)	0.1
Disease Duration (years)	5.3 (3.4)	6.2 (4.6)	0.6
LEDD (mg/day)	544 (198)	583 (191)	0.7
H&Y	2 (2–3)	2 (2–3)	0.4

Legend: CPT, conventional physical therapy; ERT, experimental robotic therapy; H&Y, Hoehn–Yahr stage determined in the “on” phase; LEDD, levodopa equivalent daily dose.

**Table 2 jcm-11-02543-t002:** Outcome measure scores at T0 (baseline) and T1 (post-treatment) in the ERT (*n* = 15; * *n* = 12) and CPT (*n* = 9) group. Data are reported as mean (SD) or median (interquartile range), and *p*-value of within-group (using *t*-test or Wilcoxon test) and between-group comparison (using *t*-test or Mann–Whitney test).

		T0	T1	Within-Group Comparison	Between-Group Comparison
9HPT	ERT	42.2 (17)	34.1 (14)	0.006	0.004
	CPT	35.1 (6.8)	31.4 (5.4)	0.9	
UPDRS-III	ERT	28 (23–33)	21 (16–26)	0.06	0.5
	CPT	37 (31.5–41)	32 (23.25;40)	0.9	
P-NRS	ERT	2.5 (0.5–3.5)	1.1 (0.3–1.8)	0.007	0.9
	CPT	4 (3–5)	1 (0–1.5)	0.01	
MI-UE	ERT	72 (65–80) *	89 (83–94) *	0.04	0.0001
	CPT	77 (73.25–82)	82 (79.25;88.5)	0.8	
FIM	ERT	104 (98–109) *	110 (105–115) *	0.6	0.6
	CPT	100 (99–103)	101 (100–106)	0.9	
FMA-UE	ERT	48 (45–52)	53 (5–56)	0.007	0.009
	CPT	53 (51–55)	56 (52.5–59.5)	0.9	

Legend: 9HPT, nine-hole peg test; UPDRS, Unified Parkinson’s Disease Rating Scale; P-NRS, numerical rating scale of pain; MI-UE, Motricity Index for Upper Extremity; FIM, Functional Independence Measure; FMA-UE, Fugl-Meyer Assessment for Upper Extremity.

**Table 3 jcm-11-02543-t003:** Affected side assessment at T0 (baseline) and T1 (post-treatment) in the ERT (*n* = 15; * *n* = 12) and CPT (*n* = 9) group. Data are reported as mean (SD) or median (interquartile range), and *p*-value of within-group (using *t*-test or Wilcoxon test) and between-group comparison (using *t*-test or Mann–Whitney test).

		T0	T1	Within-Group Comparison	Between-Group Comparison
9HPT	ERT	42.2 (17.3) *	34.1 (13.9)	0.001	0.7
	CPT	35.1 (6.8)	31.4 (5.4)	0.003	
UPDRS-III	ERT	28 (23.25–32)	21 (15–22.5)	0.07	0.7
	CPT	28 (26.25–31)	24 (23.25–26)	0.1	
P-NRS	ERT	2.1 (1.1–4.1)	1.3 (1.2–1.4)	0.01	0.001
	CPT	1.7 (1.6–1.9)	1.5 (1.4–1.6)	0.3	
MI-UE	ERT	69 (64.25–76) *	87 (82.25–94)	0.001	0.002
	CPT	84 (78–91.5)	82 (78.25–90.5)	0.4	
FIM	ERT	103 (100.25–106.5) *	109 (104.25–112.5)	0.0001	0.4
	CPT	86 (82.25–90)	122 (118.25–124)	0.0001	
FMA-UE	ERT	48 (45.25–51)	52 (50.25–54)	0.001	0.008
	CPT	56 (50–57.5)	57 (52.26–59)	0.08	

Legend: 9HPT, nine-hole peg test; UPDRS, Unified Parkinson’s Disease Rating Scale; P-NRS, numerical rating scale of pain; MI-UE, Motricity Index for Upper Extremity; FIM, Functional Independence Measure; FMA-UE, Fugl-Meyer Assessment for Upper Extremity.

## Data Availability

Data could be requested on demand to the corresponding author.

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
