# Peer review of "Improving Upper Extremity Bradykinesia in Parkinson’s Disease: A Randomized Clinical Trial on the Use of Gravity-Supporting Exoskeletons"

_jcm, 2022, doi:10.3390/jcm11092543_

Round 1

Reviewer 1 Report

The paper “Improving Upper Extremity Bradykinesia in Parkinson’s disease: a Randomized Clinical Trial on the Use of Gravity-Supporting Exoskeletons” describes a randomized controlled trial for Parkinson disease based on a commercial exoskeleton device. Results indicate that favorable effects can be achieved with the exoskeletal therapy.
The paper is in general clearly written and understandable. even if the enrolled subjects are less than the original planned number, limiting the impact of the results.

Authors should improve the quality of the Figure 1 as it is barely readable (is the consort or prisma guidelines? this should be in the caption). The “give reasons” should be removed too from the graph.

I think that answering in more detail to the following questions would improve the quality of the presentation and the conveyed message:

Is there any assumption of the rationale of the study indicating that robotic mobilization should be beneficial to Parkinson patients?

How do you explain your favorable results at neural level?

Is the benefit only due to the dose of the treatment? Relatedly, the dose of treatment when comparing the two groups were the same. You mean the same number of movements, or the same global treatment time?

Can you please clarify “secs” at line 119?

Author Response

  1. Authors should improve the quality of the Figure 1 as it is barely readable (is the consort or prisma guidelines? this should be in the caption). The “give reasons” should be removed too from the graph.

We thank for the comment and the good suggestion. Based on it, we improve the figure

I think that answering in more detail to the following questions would improve the quality of the presentation and the conveyed message:

  1. Is there any assumption of the rationale of the study indicating that robotic mobilization should be beneficial to Parkinson patients?

Thank you very much for your very interesting observation. We added these details on rationale in the introduction.

  1. How do you explain your favorable results at neural level?

We have explained that the intensive, repetitive and task oriented (also thanks to the 2D VR) may have boosted neural plasticity. Now, we have also added on the role of cerebellum in potentiating motor learning, as previously demonstrated in patients with PD receiving Gait training with music.

  1. Is the benefit only due to the dose of the treatment? Relatedly, the dose of treatment when comparing the two groups were the same. You mean the same number of movements, or the same global treatment time?

We appreciate the reviewer for the observation and clarification. We performed the same global treatment time for both groups whose improvement depended on the intensive, repetitive, assisted-as- needed, and task-oriented motor practice provided by the mechanical device [Calabrò RS, Naro A, Russo M, et al. The role of virtual reality in improving motor performance as revealed by EEG: a ran-domized clinical trial. J Neuroeng Rehabil. 2017;14(1):53. doi:10.1186/s12984-017-0268-4].

  1. Can you please clarify “secs” at line 119?

Corrected

Reviewer 2 Report

This is a well-written manuscript with an important clinical message, and should be of great interest to the journal  Improving Upper Extremity Bradykinesia in Parkinson’s disease and the Use of Gravity-Supporting Exoskeletons  is very important in order to help readers about a better knowledge.

The topic of this study is trend, so it may be helpful to the readers.

On the other Use of Gravity-Supporting Exoskeletons  in PD’s could reduce the complications of comorbidities. And could contribute to increase general health status

Introduction section is deep enough with and adequate focus that may help readers to improve knowledge about the topic. However authors should improve the stay of art, for example including references to gait and parkinson’s disease suggest to include this references include in the attached to complet this requeriment

DOI: 10.3390/brainsci10020069 

methods section is right written

Results section is enough clearly showed enough number of figures and tables that help to achieve and  understanding of this analysis. Morever the statistics analyse could be improved to increase the scientific evidence level

Discussion section is well structured with different sections. Authors manage well the discussion leading a good comparison with the showed references.

However authors should compare their results with other prior researcher about health related quality of life in chronic disorders

I suggest to include the following references to complete this requeriment

DOI: 10.1038/s41598-021-93902-5

Author Response

  1. Introduction section is deep enough with and adequate focus that may help readers to improve knowledge about the topic. However, authors should improve the stay of art, for example including references to gait and parkinson’s disease suggest to include this references include in the attached to complet this requeriment

We thank for the comment and the good recommendation. We added this information with the reference, as suggested:

Although we have not specifically investigated quality of life, it is possible that the improvement in disability burden has also improved patients’ health-related quality of life, as suggested by other chronic disease (López-López D, Pérez-Ríos M, Ruano-Ravina A, et al. Impact of quality of life re-lated to foot problems: a case-control study. Sci Rep. 2021;11(1):14515. Published 2021 Jul 15. doi:10.1038/s41598-021-93902-5). (DOI: 10.3390/brainsci10020069).

  1. methods section is right written. Results section is enough clearly showed enough number of figures and tables that help to achieve and understanding of this analysis. Moreover the statistics analyses could be improved to increase the scientific evidence

Although statistics could improve scientific evidence, it is not possible to perform other analysis because of the small sample size (as also stated by a consultant statistician). We added this issue in the limitation.

  1. Discussion section is well structured with different sections. Authors manage well the discussion leading a good comparison with the showed references. However, authors should compare their results with other prior researcher about health-related quality of life in chronic disorders; I suggest to include the following references to complete this requirement.

Thank you for your good suggestion. We added and commented the reference as suggested (López-López D, Pérez-Ríos M, Ruano-Ravina A, et al. Impact of quality of life related to foot problems: a case-control study. Sci Rep. 2021;11(1):14515. Published 2021 Jul 15. doi:10.1038/s41598-021-93902-5).

Round 2

Reviewer 2 Report

authors have adressed all my requirement in the correct way